# Genome-Wide Identification and Comparative Analysis of Myosin Gene Family in Four Major Cotton Species

**DOI:** 10.3390/genes11070731

**Published:** 2020-06-30

**Authors:** Chenhui Ma, Zibo Zhao, Na Wang, Muhammad Tehseen Azhar, Xiongming Du

**Affiliations:** 1State Key Laboratory of Cotton Biology, Institute of Cotton Research of the Chinese Academy of Agricultural Sciences, Anyang 455000, China; charlesjack7@163.com (C.M.); zzb18039221701@163.com (Z.Z.); 15517960172@163.com (N.W.); 2Department of Plant Breeding and Genetics, University of Agriculture, Faisalabad 38000, Pakistan; tehseenazhar@gmail.com

**Keywords:** cotton, Myosin gene family, evolution, expression pattern

## Abstract

Myosin protein as a molecular motor, binding with Actin, plays a significant role in various physiological activities such as cell division, movement, migration, and morphology; however, there are only a few studies on plant Myosin gene family, particularly in cotton. A total of 114 Myosin genes were found in *Gossypium hirsutum*, *Gossypium barbadense*, *Gossypium raimondii*, and *Gossypium arboreum*. All Myosins could be grouped into six groups, and for each group of these genes, similar gene structures are found. Study of evolution suggested that the whole genome duplications event occurring about 13–20 MYA (millions of years ago) is the key explanation for Myosins expanse in cotton. Cis-element and qPCR analysis revealed that plant hormones such as abscisic acid, methyl jasmonate, and salicylic acid can control the expression of Myosins. This research provides useful information on the function of Myosin genes in regulating plant growth, production, and fiber elongation for further studies.

## 1. Introduction

Myosins are molecular motors that can convert adenosine triphosphate (ATP) released chemical energy into physical motion. They can carry various intracellular cargos including membranous, organelles, protein complexes, and mRNAs. Myosins usually have three parts, ATP-dependent motor domain was the first part and located in the top of protein. A neck domain with IQ motifs in different quantities which can bind protein-like calmodulin or light chain was in the middle, and a tail dominant may bind different “cargo” was in the end [1]. There were at least 35 Myosin classes in all eukaryotes, but there are only two Myosin classes which are VIII and XI in plants [2]. As far as Class VIII Myosins are concerned, these contain C-terminal coiled–coil regions and long N-terminal extension. Class XI like Myosin Class V has six IQ patterns with an extension of coiled coil region as well as a DIL domain. In *Arabidopsis*, there are 13 isoforms of Myosin XI (XI1, XI-2, XI-A, XI-B, XI-C, XI-D, XI-E, XI-F, XI-G, XI-H, XI-I, XI-J, and XIK) and two isoforms of Myosin VIII including VIII-A, VIII-B, ATM1, and ATM2 [3]. Rice has 12 Myosin XI and 2 Myosin VIII [4]. In *Zea mays*, there are 11 Myosin XI and 3 Myosin VIII [5].

Myosin XI genes typically have much more members than class VIII, so further research on Myosin XI in cytoplasmic streaming, contribution of actin, organization and dynamic transport, and function diversity were published [6]. The treatment experiment with an actin inhibitor (cytochalasin B) shows that actin filament produces motivation [7]. Purified tobacco 175-kDa Myosin XI is shown to have a cytoplasmic streaming velocity of Myosin XI. Tominaga et al. uses *CharaMyo XI* as their quickest replacement for *Arabidopsis* Myosin XI-2 to speed up the growth of cytoplasm and plants [8]. All these results suggest that Myosin XI is the driving force for cytoplasmic streaming and one of the plant growth regulators. Several studies have shown that F-actin’s Myosin XI is a regulator. In *Hydrocharis dubia*, 2,3-butanedione monoxime (BDM), which is considered as a Myosin ATP hydrolysis inhibitor, can be classified as F-actin disorder [9]. In *Arabidopsis* xi-2 xi-k xi-1 knockouts, the level of F-actin sever was decreased by two times [10]. Myosin XI also plays a significant role in diffusing or polarizing plant cell growth patterns. A remarkable spreading defect of diffuse growth was identified by the combination of *ATXI1-1*, *ATXI1-2* knockout mutants [11]. Myosin XI also affects root hair—cylindrical cells that are exceptionally long and polarized [12]. Unlike Myosin class XI, features of class VIII are less well known [13]. An *ATM1* can act as a sensor-generator in a variety of intra-cellular structures, such as cell cortex, newly shaped cell wall, plasmodesmata, and plastids [14]. According to Takeshi Haraguchi et al.’s work in tobacco, the plasmodesma and the endosomal tracer *FM4-64* were co-located with *ATM1* [15]. Lots of findings show that Myosin VIII proteins may be associated with protonemal patterning of endocytosis, cytokinesis, and plasmodesmal.

Myosin, as an important, conserved, and widely distribution gene family in eukaryotes, was easy to track the path of its gene family expansion [16]. *Anborella trichipoda*, which has five Myosins VIII, including 11A’, 11C’, 11E’, 11G’, and 11H’ was considered as a sister to the ancestors of the eudicots as well as monocots and assumed to be diverged followed by two ancient whole genome duplications (WGDs), which occurred in the evolution of angiosperms [17]. Thus, all five subtypes undergo the 192 and 319 million years ago (MYA) WGD event, but they have no evidence of the hexaploid event at the origin of the core eudicots. In addition to 11A, 11C, 11E, 11G, and 11H, eudicots have three extra Myosin XI subtypes 11B, 11D, and 11F than monocots. The γ hexaploidy of core eudicots may be the cause of the increase of that. In some single species, loss of 11F as well as 11H occurred just like the *Eucalyptus grandis* Myosin-11H lost. In addition, it is assumed that Myosin VIIIA and VIIB would be possibly resulted from an ancient time duplication of a single gene or genome within common ancestor of whole extant angiosperms [16].

Cotton is an economic and fibrous crop of global significance. Actin and actin-binding protein were found to play an important role in the growth [18] and elongation [19] of plants. Cotton Myosin should be essential for the growth of fiber as an interacting protein with actin, but relatively few studies exist. We described 114 Myosins, including two diploids and two tetraploids, among four cotton species. Phylogenetic analysis as well as chromosomal distribution, gene duplication through collinearity analysis, promoter cis-elements, Ka/Ks ratios, and gene structure were also predicted. The expression profiles of the families of Myosins were determined. This research is required to test Myosin’s physiological and biochemical functions.

## 2. Materials and Methods

### 2.1. Identification of Myosins in Gossypium hirsutum, Gossypium barbadense, Gossypium raimondii and Gossypium arboretum

*Gossypium hirsutum*, *Gossypium barbadense*, *Gossypium raimondii*, as well as *Gossypium arboreum* genomic data, annotation, and expression data were downloaded from the CottonFGD (http:/cottonfgd.org/) website [20]. We checked for a sequence that includes Myosin_head domain (PF00063) and IQ domain (PF00612) for the identification of Myosin gene family members by using HMMER3.0 (e-value 10${}^{-5}$). The nominee was confirmed with NCBI-CDD (http:/www.ncbi.nlm.nih.gov/cdd) [21].

### 2.2. The Analysis of Myosin Proteins Characteristic

Protein Analysis Characteristic Online ExPASy program (https:/web.expasy.org/computepi/) estimated an amino acid iso-electric point and molecular weight. Targetp1.1 (http:/www.cbs.dtu.dk/services/TargetP/) Subcellular location for online applications. Tbtools were used for the study of chromosome position [22]. PlantCare’s online programme (http:/bioinformatics.psb.ugent.be/webtools/plantcare) was used to predict the cis-elements within the promoter.

### 2.3. Phylogenetic Tree Construction and Evolution Analysis

The website (https:/www.cymobase.org/cymobase) was used to download all of the Myosin sequences of *Arabidopsis thaliana*, whereas for *Amborella trichopoda*, *Drosophilia melanogast* Myosin V, and *Homo sapiens* Myosin X, we used NCBI. To create a phylogenetic tree, ClustalW aligns sequences and MEGA7 uses neighboring (NJ) and maximum similarity (ML) cross-species and intraspecific tree with 1000 bootstrap replicates, respectively [23]. To classify homologous genes, collinearity and Ka/Ks, Tbtools, MCscanx, and Dnasp were used [24].

### 2.4. Plant Material, Treatment and qRT-PCR Analysis

We have used TM-1 (*Gossypium hirsutum*) for the purpose of RNA extraction as well as qRT-PCR. In order to perform plant hormones treatment, seeds were germinated in sand and then transferred to hydroponic solution where Hoagland’s was applied at the early stage. Different concentrations of methyl jasmonate (MeJA) (100 μmol/L), gibberellic acid (GA) (100 μmol/L), salicylic acid (SA) (100 μmol/L), and abscisic acid (ABA) (100 μmol/L) were used in order to treat cotton seedlings at the four true leaves stage. Samples of leaf, root, and stem apex were collected at 12 h following treatment. Collected samples were placed in liquid nitrogen and stored at −80${}^{{}^{\circ}}$C. Subsequent RNA extraction was carried out using an RNAprep Pure Plant Kit (TIANGEN Biotech, BeiJing, China). The extracted RNA was used for reverse transcription and 1 μg of total RNA was used to be transcribed into cDNA by using One-Step RT-PCR Kit (Novoprotein Scientific, BeiJing, China). The qPCR was performed by using ChamQ Universal SYBR qPCR Master Mix (Vazyme Biotech, NanJing, China). LightCycler 480 by Roche Diagnostics GmbH, Mannheim, Germany was used to perform PCR amplification. Detailed Primer information used in the current study is given in Appendix A. The GhACT4 was chosen as the housekeeping gene in website (http://icg.big.ac.cn/index.php/Gossypium_hirsutum).

## 3. Results

### 3.1. Identification of Myosin Genes in Four Cotton Species

Through the four cotton species viz. *Gossypium hirsutum*, *Gossypium barbadense*, *Gossypium raimondii*, and *Gossypium arboreum*, we detected a total of 114 Myosins proteins by confirming that they contain Myosin head and IQ domain. For diploid cotton, the family members of the 19 and 20 Myosin genes belong, respectively, to *Gossypium raimondii* and *Gossypium arboretum*. In *Gossypium hirsutum* the subgenome At and Dt both have 19 genes. The 19 genes of *Gossypium barbadense* At and Dt have 17. The number of Myosin genes in each tetraploid cotton is approximately twice that of the A or D donor genome. The findings show that after ployploidization, there is almost no loss of the Myosin cotton gene.

The length of all 114 protein Myosins were between 900 and 2034 AA (amino acid), however that gene length is widely distributed between 7360 and 37373 bp (base pair). *GOBAR_DD34721.1* even has 52 exons and all average number of exons were 32.86. Most of the Myosin genes’ isoelectric points were higher than seven, with just 17 Myosins being amino acid. For all Myosin genes, the great average of hydropathy is less than 0, which means all genes are hydrophilic proteins. A total of 14 Myosins are located in chloroplast for cell localization, and 21 proteins contain signal peptide which means they are protein secreted. Interestingly, only three *GOBAR_AA27044.1*, *GOBAR_AA19615.1*, and *GOBAR_DD13861.1* island cotton gene possess the mitochondrial targeting peptide. Appendix A displays all the essential findings of bioinformatics research.

### 3.2. Phylogenetic, Conserved Domain/Motifs, and Gene Structures Analysis in Four Cotton Species

We used preserved phylogenetic, domain/motifs, and gene structure analysis to divide these genes into six classes to identify the all 114 Myosins from four cotton species. Only two species of Myosin, XI and VIII, existed in cotton. Classes one to four (77 Myosins), and five to six (37 Myosins), belong, respectively, to Myosin XI and Myosin VIII (Appendix A). In all cotton Myosins, we classified 20 motifs. Genes identified in the same category have the same form of conservative non-family domain as DIL, IncA, and Laminin II. Getting this domineering can affect its operation. In N-terminus and motif-13 in the C-terminus, group VI genes essentially just have a Myosin head and IQ domain. Some Myosins like *Gh_D08G2035.1*, *Gorai.004G279700.1*, and *Ga08G2238.1* miss the C-terminus motif and dominant.

As noted above, the number of exons and introns varies widely across gene families, and there is no correlation between protein and gene length. *Gh_A03G2151.1* has only 21 exons (the least one), but the length of the protein was 11550 AA in the middle level. Other gene length characteristics and classification findings were clear, except for certain genes such as *Gh_A10G1061.1* and *Gh_D09G0526.1*, which have exceptionally long gene lengths. The same subfamily has been found to have identical motif features, gene length as well as exon–intron structure confirming the findings of the evolutionary classification as shown in Figure 1.

### 3.3. Gene Duplication, Collinearity, and Evolutionary Analysis

We used MCscanX to find colinear blocks with a blast threshold of 1e${}^{-5}$. Gene pairs that are in one pair of collinear blocks were considered to be homology genes that were found in the same chromosome that were separated by less than five genes, which we considered to be tandem repeats. Finally, we found 252 paralogs and 102 orthologists. Only *Gossypium arboretum* has one pair of tandem repeats (*Ga05G4016.1* and *Ga05G4019.1*). The Ka/Ks ratio for all homology gene pairs was less than 1, with the exception of the nine paralogic pairs (Appendix A).

We used *Amborella trichopoda* Myosins to create a phylogenetic tree with *Gossypium raimondii* and *Gossypium arboreum*, respectively, to reliably track the evolution. Every diploid cotton tree with *Amborella trichopoda* Myosins can both divide into seven clusters in accordance with the existing results of the classification. We regard the cotton Myosins closest the *Amborella trichopoda* Myosins as the original gene, and then measured the time of divergence between the original gene and the same cluster gene with the eq. $T=Ks/2\lambda \times 10^{-6}$ MYA (million years ago), (approximate value for clock-like rate $\lambda =1.5\times 10^{-8}$ years for cotton) [25]. The results shows that the pathway of Myosin gene expanse in *Gossypium raimondii* and *Gossypium arboreum* can be divided into six stages, which is consistent with important genomic changes in cotton evolution. As we all know, *Gossypium hirsutum* and *Gossypium barbadense* are made from two diploids, so we developed collinearity plots for two species of diploid and tetraploid cotton, respectively. The results show that only six loci of *Gossypium arboretum* have homology which means that *Gossypium raimondii* has more contribution in the formation of tetraploids showing in Figure 2.

### 3.4. Cis-Element Analysis, Expression Pattern, and Hormone Treatment Expression Analysis

We examined the pattern of expression in various organelles of the *Gossypium hirsutum* and *Gossypium barbadense* Myosin genes (Appendix A). The findings show that most half of the Myosin genes are low expression in all organizations and other genes, such as *Gh_A3G2151.1* and *Gobar_AA38264.1* and high expression genes, are widely expressed in different sections. According to the results of expression clustering, genes that are primarily expressed in anther tissues are also strongly expressed in flowers as well as fiber.

The promoter analysis shows that all Myosins have 42 kinds of cis-element in 2000 bp up-stream area, seven different light responsive element, and five elements about plant hormone response (Appendix A). We treated TM-1 with GA, MeJA, ABA, and SA at a concentration of 100 um/L in ten Myosins from *Gossypium hirsutum*, which were randomly picked. SA only upregulates the expression of Myosins in the roots, and negatively regulates in the stems and leaves. However, Myosin in the leaves and stems is mostly induced by MeJA and ABA. The regulation effect of GA is not obvious. Different Myosins were regulated by different hormones, both positive and negative, showing in Figure 3.

## 4. Discussion

Myosins, as a motor protein, which can converts chemical energy into physical movement, are widely distributed in eukaryotes [6], which can be typically classified into 35 classes, where only class VIII and class XI exist in plants [25]. Myosin, the binding protein of actin, which is confirmed to be involved with fiber elongation, still has no systematic analyses of the cotton Myosin gene family. In the current study, we investigated the Myosin gene and analyzed the evolution as well as the expression pattern after hormone treatment.

We found 114 Myosins in all four cotton species with the principle that sequences both contain Myosin_head domain (PF00063) and IQ domain (PF00612). Cotton Myosin group five and six lacks DIL domain, which consists of maybe 5, 11, and 19 motifs. Whereas, Myosin-VIII is categorized as a slow motor protein, Myosins-XI have been categorized as the fastest motor proteins [26]. Myosin XI were also assumed to be necessary in the transportation of different organelles like Golgi sacks, peroxisomes, and mitochondria, whilst Myosin-VIII was found to be involved across endocytosis as well as plasmodesmata targeting [27]. Lacking a DIL domain, the “cargo” binding area may be related to a different function between Myosin XI and Myosin VIII. Interestingly, cotton Myosins have lots of exon, *GOBAR_DD34721.1* even have 52 exons, which means alternative splicing can usually occur. The alternative splicing offers the possibility of regulatory functions performed through Myosin partners selectively binding. For example, DIL-less isoforms are likely to only bind F-actin, modulating transportation capacity, Myo1b can also alternatively spliced in its light chain binding domain (LCBD) to produce different lengths lever arms with the different of magnitude of tension sensitivity [28].

We all know *Gossypium hirsutum* (38 Myosin genes) and *Gossypium barbadense* (37 Myosin genes) were a combination of *Gossypium raimondii* (19 Myosin genes) and *Gossypium arboretum* (20 Myosin genes) [29]. The number of Myosin gene in tetraploid cotton is almost double that of diploid cotton [30]. This result shows that during the formation of tetraploid cotton, members of the Myosin gene family may not experience loss and expansion. There are 2 Myosin class XI and 12 Myosin VIII in *Oryza sativa cv.* [31], 4 Myosin class XI, 13 Myosin VIII in *Arabidopsis Thaliana* [32], 7 Myosin class XI, and 22 Myosin VIII in *Glycine max* [33]. The number of other plant Myosin gene is almost consistent with the chromosome ploidy, which means that the expansion of Myosins may only follow the differentiation of the species, and its number is consistent with the size of the chromosome. The *Amborella trichopoda*, as the earliest differentiated species from the angiosperm, have seven Myosin genes viz. 8A, 8B, 11A, 11C, 11E, 11G, and 11H. During species evolution, Myosin 11F as well as 11H may possibly be lost within some species during the evolution [16]. *Gossypium raimondii* and *Gossypium arboretum* have already lost the 11H class member.

Expansion of most gene family members depends on whole genome duplications (WGD) events. *Gossypium raimondii* and *Gossypium arboretum* cotton Myosin members of the increase process can be divided into six stages, consistent with the evolution of cotton species [3], which means Myosins were almost generated through the big evolution events like WGD or cotton and cocoa differentiation. In diploid cotton, the eight times expanse event was the cause of the 13–20 MYA (million years ago) WGD event after cotton and cocoa differentiation, and all expanse event times were 24, so that this WGD event played an important part in the expansion of the Myosin family. Only one tandem happened between *Ga05G4016.1* and *Ga05G4016.1*, and maybe that is why *Gossypium arboretum* has one more Myosin member than *Gossypium raimondii*. More than 95% of the Ka/Ks ratio of the homologous gene pair is less than one, indicating negative selection during evolution. The results showed that most Myosin genes are slowly evolving. For tetraploid cotton, we can trace most of the homologous genes of the Myosin gene from two diploid donors; however, it is interesting that *Gossypium arboretum* seems to provide more Myosins during the formation of tetraploid cotton. During the formation of tetraploid cotton, there are lots of chromosomal rearrangement, transposition, and gene loss, which may influence the collinearity detection of micro-regions [3]. All evolutionary results indicate that the Myosin family is very strict and conservative during the expansion of its species, which may be related to maintaining important functions, whether in plants or animals such as cell differentiation [34], cytoplasmic streaming, and actin organization [35].

The expression pattern shows that half of the Myosin genes are low expressed, which shows that most Myosin are redundant [36], and some experiments have confirmed this [37]. The cis-element and qRT-PCR results shows that lots of plant hormone can have both positively and negatively regulated the Myosin gene. The moss’ Loss of all five Myosin VIII genes mutant can cause growth and development to be inhibited, and most of the morphological defects can be saved by low concentrations of auxin. These results indicated that Myosin VIII was involved in maintaining proper hormone balance, at least in moss [6]. Actin or actin-related genes like *ACTIN1* [38], *GhPFN2* [39], and *GhVLN4* [40] have been found to play an important role in the development of cotton fiber. As a binding protein of actin, although no direct reported evidence indicates that Myosin can influence fiber elongation [41], the Myosin influence on actin assembly and plant development has been extensively studied [42]. Therefore, more experimental evidence is needed to confirm its role in cotton fiber development in the future.

## 5. Conclusions

In this study, we identified Myosin gene family members in four cotton species *G. hirsutum*, *G. barbadense*, *G. raimondii*, and *G. arboreum*. We have conducted a comprehensive phylogenetic, evolutionary, and hormone response analysis. The results revealed that Myosin is conservative, and the WGD event contributed greatly to the expansion of Myosin in diploid cotton. Myosin can be regulated by a variety of hormones, and its role in cotton fiber development needs further verification.

## Figures and Tables

**Figure 1 genes-11-00731-f001:**
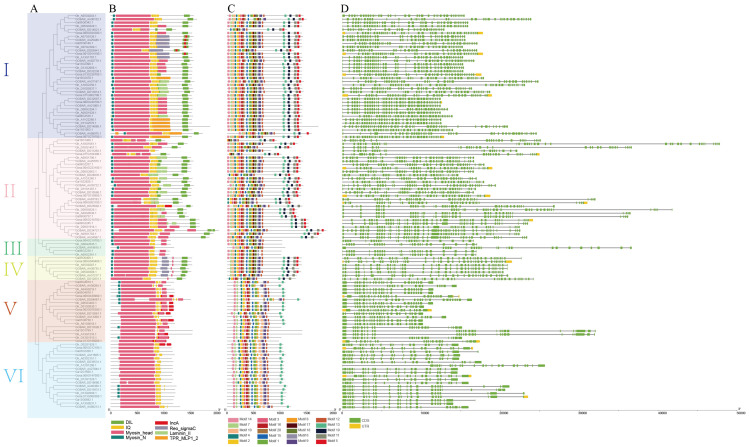
Comparison of the gene structures, domin, and motif distribution pattern of Myosin genes in four cotton species. (**A**) The phylogenetic tree of Myosin genes. (**B**) The distribution pattern of predicted motifs in the Myosin genes. (**C**) The distribution pattern of predicted domain in the Myosin genes. (**D**) The position of exons and introns within Myosin genes. Exons, UTR, and introns are shown by green boxes, yellow boxes, and black lines, respectively.

**Figure 2 genes-11-00731-f002:**
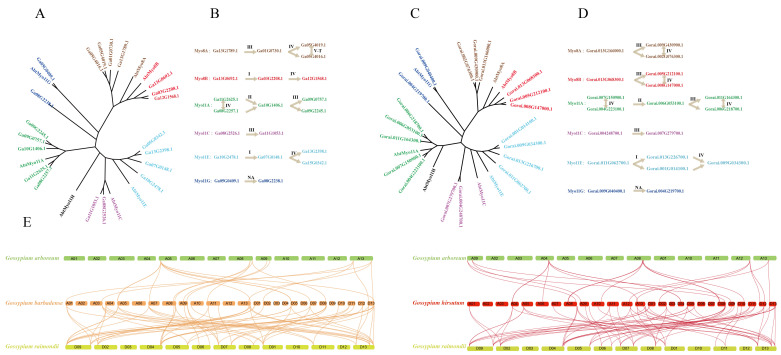
Gene Duplication, collinearity, and evolutionary analysis. (**A**) The phylogenetic tree between *Amborella trichopoda* and *Gossypium arboreum* Myosins. (**B**) The *Gossypium arboreum* Myosins expansion pathway. The arrows point to new genes generated by gene expansion events and the character above the arrow I, II, III, IV, V, V-T, and NA means the ancestors of cocoa and cotton differentiated from *Arabidopsis* at about 93 MYA (million years ago), the differentiation time between cotton and cocoa at about 33 MYA (million years ago), ancestral cotton WGD event at 13∼20 MYA (million years ago), the differentiation time between *Gossypium arboreum* and *Gossypium raimondi* at about 2∼13 MYA (million years ago), tetraploid cotton formed in 1.5 MYA (million years ago), tandem event after tetraploid cotton formation, and unknown event, respectively. (**C**) The phylogenetic tree between *Amborella trichopoda* and *Gossypium raimondi* Myosins. (**D**) The *Gossypium raimondii* Myosins expansion pathway, the meaning of arrows and character was same as described in (**B**). (**E**) The collinear relationship of homologous genes between *Gossypium arboreum*, *Gossypium raimondii*, and *Gossypium barbadense*, *Gossypium hirsutum*.

**Figure 3 genes-11-00731-f003:**
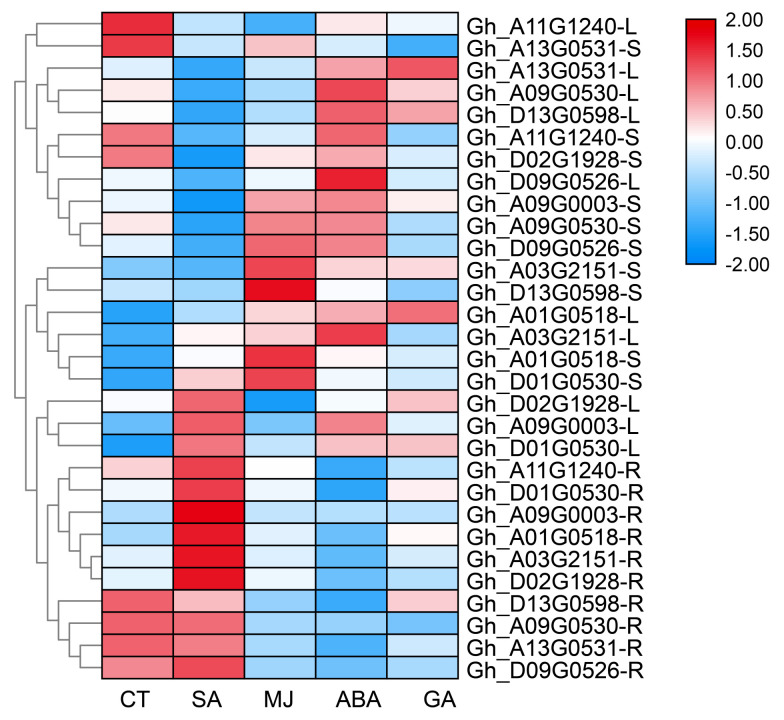
The expression of ten *Gossypium hirsutum* Myosins. CT, SA, MJ, ABA, and GA mean control and different hormone treatment like salicylic acid, methyl jasmonate, abscisic acid, and gibberellic acid. The letters S, L and R after the gene name represented the root, leaf and stem organs of the cotton.

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
