# Peer review of "Genome-Wide Identification and Comparative Analysis of Myosin Gene Family in Four Major Cotton Species"

_genes, 2020, doi:10.3390/genes11070731_

Round 1

Reviewer 1 Report

General comments:

The authors identified Myosin gene family in four cotton species.  They conducted comprehensive phylogenetic, evolutionary, and hormone response analysis. They concluded that Myosin is conservative and the whole genome duplication event contributed greatly to the expansion of Myosin in diploid cotton. Moreover, this family of genes could be regulated by hormones.   

In general, the manuscript is well written and presented. However, there are a few errors that the authors must change.

Some comments are provided below regarding some suggestions about how the manuscript might be improved.

Specific comments

Title: Change varieties for species. Gossypium hirsutum, Gossypium barbadense, Gossypium raimondii and Gossypium arboreum are not varieties are different species.

Abstract

Line 8. Spell all the word Methyl jasmonate (MejA).

Introduction

Line 15. Extra space before “They”

Line 16. It is domin or domain?

Line 17. Should be domain and not domin change in whole manuscript.

Line 17. What is the connection of this sentence with the previous one?

Line 18. Change “There are” for “It has”

Line 19. Change 2 for two.

Line 30. Arabidopsis in italic font

Line 32. Add dubia after Hydrocharis

Line 47. Space after WGDs

Line 49. The first time in the article that you write MYA (million years ago), you should write all the words.

Line 53. Grandis has to be in lowercase

Line 53. Add reference.

Line 57. Space after [18].

Lines 58, 62, 102, and 215. The word myosin is with lowercase in these lines, however, in the whole document you have with a capital letter. You should change all to the capital letter or all lowercase; be consistent.

Materials and Methods

Line 80. trichopoda, melangast and sapiens must be in lowercase (mistake in all the document)

Line 95. There is a space extra after China).

Line 98. There is not available the supplementary files.

Line 98. What is the reference housekeeping gene that you use for normalizing the gene expression in this experiment?

Results:

Line 100. Add species after cotton

Line 101. There are not varieties there are different species.

Line 103. There is (_) extra.

Line 108. This sentence is not belonging in the result section. You should change to the discussion and add the reference.

Line 112-113. In the results section, you should just report the results and not the explanation, the explanation must be in the discussion.

Line 114. A simple rule for using numbers in writing is that small numbers ranging from one to ten (or one to nine, depending on the style guide) should generally be spelled out. Thus, in all your document you should spell out the number like seven.

Line 120. Domain and not domin. Add species after cotton.

Line 123. Spell out the numbers.

Figure 1. Add species after cotton. Change domain for domin.

Line 139. Spell out the number 5.

Figure 2. Mya has to be all in capital letters as you put inline 49.

Lines 143 and 145. Trichopoda has to be in lowercase and spell out the number 7.

Figure 3. The letters SA, MJ, ABA, and GA should put the name of the hormones. The title there is a typo in control.

Line 163. Spell out the numbers 7 and 5.

Line 165. Gossypium hirsutum in Italic font.

Discussion:

Line 176. Erase finally; add species after cotton.

Line 177. Correct the word domin

Line 179. Add a space before the reference.

Line 182 Correct the word domin

Line 190. Add a space before the second reference

Line 196. trichopoda, must be in lowercase

Line 202. Gossypium arboreum must be in italic font.

Line 208. Gossypium arboreum must be in italic font

Line 209. Gossypium raimondii must be in italic font

Line 212. Gossypium arboreum must be in italic font

Line 219. Add a space after “expressed”

Line 224. Add a space before the reference

Conclusions:

Line 228. Change varieties for species.

References:

No comments.

Reviewer 2 Report

The manuscript provides a fairly robust information, but it needs English editing and also the results should be discussed better in discussion section.

Some  suggestions:

- Use Myosin in all the manuscript not myosin as in lines 9, 58, 62

- In line 86, 165, 208, 209,212, 316, 324,335  Gossypium hirsutum should be in italic

- Line 101, 228 Gossypium hirsutum, Gossypium barbadense, Gossypium raimondii and Gossypium arboretum are not varieties but are species
